# Multi-Scale Digital Image Correlation Analysis of In Situ Deformation of Open-Cell Porous Ultra-High Molecular Weight Polyethylene Foam

**DOI:** 10.3390/polym12112607

**Published:** 2020-11-06

**Authors:** Eugene S. Statnik, Codrutza Dragu, Cyril Besnard, Alexander J.G. Lunt, Alexey I. Salimon, Aleksey Maksimkin, Alexander M. Korsunsky

**Affiliations:** 1HSM Lab, Center for Energy Science and Technology, Skoltech, 121205 Moscow, Russia; a.salimon@skoltech.ru (A.I.S.); Alexander.Korsunsky@eng.ox.ac.uk (A.M.K.); 2MBLEM, Department of Engineering Science, University of Oxford, Oxford OX1 3PJ, UK; codrutza-maria.dragu@trinity.ox.ac.uk (C.D.); cyril.besnard@eng.ox.ac.uk (C.B.); 3Department of Mechanical Engineering, University of Bath, Bath BA2 7AY, UK; A.J.G.Lunt@bath.ac.uk; 4Center for Composite Materials, NUST MISiS, 119049 Moscow, Russia; aleksey_maksimkin@mail.ru; 5Institute of Physiologically Active Compounds of the Russian Academy of Sciences, 142432 Chernogolovka, Russia

**Keywords:** SEM-DIC, Ncorr, porous UHMWPE, Deben Microtest, *Avizo*, tomography

## Abstract

Porous ultra-high molecular weight polyethylene (UHMWPE) is a high-performance bioinert polymer used in cranio-facial reconstructive surgery in procedures where relatively low mechanical stresses arise. As an alternative to much stiffer and more costly polyether-ether-ketone (PEEK) polymer, UHMWPE is finding further wide applications in hierarchically structured hybrids for advanced implants mimicking cartilage, cortical and trabecular bone tissues within a single component. The mechanical behaviour of open-cell UHMWPE sponges obtained through sacrificial desalination of hot compression-moulded UHMWPE-NaCl powder mixtures shows a complex dependence on the fabrication parameters and microstructural features. In particular, similarly to other porous media, it displays significant inhomogeneity of strain that readily localises within deformation bands that govern the overall response. In this article, we report advances in the development of accurate experimental techniques for *operando* studies of the structure–performance relationship applied to the porous UHMWPE medium with pore sizes of about 250 µm that are most well-suited for live cell proliferation and fast vascularization of implants. Samples of UHMWPE sponges were subjected to *in situ* compression using a micromechanical testing device within Scanning Electron Microscope (SEM) chamber, allowing the acquisition of high-resolution image sequences for Digital Image Correlation (DIC) analysis. Special masking and image processing algorithms were developed and applied to reveal the evolution of pore size and aspect ratio. Key structural evolution and deformation localisation phenomena were identified at both macro- and micro-structural levels in the elastic and plastic regimes. The motion of pore walls was quantitatively described, and the presence and influence of strain localisation zones were revealed and analysed using DIC technique.

## 1. Introduction

Since its invention and commercialization in the 1950s, ultra-high molecular weight polyethylene (UHMWPE) has been known as a high-performance polymer successfully applied in diverse engineering systems ranging from strong ropes for naval demands and wear-resistant liners in bearings, transportation belts and heavy trucks in mines and quarries, through the lining of chemical vessels and disposable bags in bioreactors, to sophisticated products such as orthopaedic implants and replacements of bone fragments in cranio-facial reconstructive surgery, hip and knee joints. The number of scientific publications has gradually increased over the last 50 years and currently exceeds an annual rate of ~350 papers [1].

Due to the combination of being bioinert and possessing good mechanical performance, special grades and formulations of UHMWPE-based materials have been developed for biomedical usage. A much stiffer and costly but 3D-printable polymer, polyetheretherketone (PEEK) recently entered the field as a strong competitor of UHMWPE in biomedical applications, presenting several challenges for the companies engaged in the materials synthesis and end product fabrication. In contrast with PEEK, UHMWPE is shaped mainly via hot moulding, although extrusion and mechanical cutting has proved the ability to form strong self-reinforced composites [2,3], architected and hierarchically structured hybrids with bioinert Titanium (Ti) [4], PEEK [5], hydroxyapatite (HAp) and collagen hybrids for advanced implants mimicking cartilage [6,7], cortical and trabecular bone tissues within a single component.

The fabrication of UHMWPE foams for biomedical applications via several techniques appears to represent the most serious development in the last decade from the materials engineering perspective. Cellular open-cell UHMWPE with the appropriate cell size of a few hundred micrometres obtained via hot sintering of loose powder or via sacrificial desalination of hot compression-moulded UHMWPE-NaCl powder mixtures is well-suited for fast vascularization, osseointegration [8], and broad interaction with live cells [9]. Close-cell UHMWPE aerogels obtained via supercritical CO_2_ extraction of a solvent [10] promise new developments in mechanical engineering (damping and thermal insulation) and smart applications when hybridized with open-cell UHMWPE sponges [11]. In contrast to bulk UHMWPE’s mechanical performance, which has been extensively studied for hip joint applications, the same for cellular UHMWPE is still to be thoroughly carried out with suitable methodological developments. For example, we recently showed [12] that significant inhomogeneity of strain distributed over the stochastic cellular structure seems to govern the overall response at static and dynamic compression, bending, and tension open-cell porous structures obtained via sacrificial desalination of hot compression-moulded UHMWPE-NaCl powder mixtures. The reported elastic and plastic deformation response could not be interpreted in terms of the classical theory of cellular solids, e.g., as described by Ashby and Gibson [13] when the cellular solids (and foamed polymers as a particular case) are considered as a set of single cells of relatively simple geometry which are elastically collapsing under mechanical load.

The issue of deformation behaviour of cellular UHMWPE—elastic, plastic, viscous, and fatigue—is of great importance for several biomedical applications for the predictive modelling of interactions with live entities such as live cells, vessels, connective and bone tissues. Overall engineering of long-lasting biomedical articles of high functionality must pass subsequent stages from conceptual design through detailed computational modelling and strength analysis to the product specification. It must rely on a consistent and experimentally validated model of static and dynamic mechanical response. This model must capture and integrate structure parameters such as cell shape (aspect), cell size distribution, and eventually wall, nod, and strut thickness. As a necessary preliminary step, the characterization (and, at first, high-resolution 2D and 3D imaging) of cellular structure morphology and its evolution at various deformation modes is required in *operando* regimes since relaxation creep events significantly affect the complete picture making *ex situ* and *post mortem* studies irrelevant.

In this article, we purposely address several methodological issues, especially the development of *operando* experimental techniques to acquire and analyse a set of high-resolution SEM images during in situ compression using a micromechanical-testing device within an electron microscope chamber. High-resolution digital 2D images are especially suitable for Digital Image Correlation (DIC) analysis facilitating the quantification and mapping of structure element movement and strain with subpixel resolution. The Digital Volume Correlation (DVC) analysis can be applied to 3D X-ray tomography images. This advanced characterization technique may not be applicable for fast processes unless a highly brilliant synchrotron X-ray beam is used to probe the samples being mechanically loaded in complex setups. That is why the comparative studies of both methods are highly desirable for reciprocal validation.

A large portion of the image of a porous structure is occupied by the empty space (pores), making DIC analysis a specifically challenging task. To tackle these issues, we directed our attention towards elaborating and refining the required processing algorithms for the segmentation of images into pores and walls and rational masking for subsequent analysis. Detailed DIC analysis leads us to the main original finding—the deformation localization phenomenon was visualized and highlighted. We discuss these deformation localization phenomena and identify them as the key structural evolution mechanisms taking place both at the macro- and micro-structural levels within the elastic and plastic regimes. Through this, we explain the disagreement of observed deformation response with the elastic Ashby–Gibson model. The lack of regularity and repeatability in the random porous structure response, e.g., the non-homogeneous collapse of pores observed during compression, is related to spontaneous strain localization. This highlights the need to introduce a statistical description of deformation inhomogeneity to capture the details of macroscale material response.

## 2. Materials and Methods

### 2.1. Porous UHMWPE Sample Preparation

Porous samples were prepared from UHMWPE powder (average molecular weight ~ 5 × 10^6^ g/mol, particle shape ~ spherical, average particle size ~ 200 μm) with the trademark GUR 4120 of Ticona GmbH production (Oberhausen, Germany) in combination with usual table salt (particle shape ~ quasi-cubic, average particle size ~ 250 μm) of LLC Salina production (Rostov, Russia) that were obtained as supplied. The structure and morphological characteristics of used powders are illustrated in Figure 1.

The porous sample preparation method included the following technological procedures: mixing, hot pressing, and a salt dissolution step according to the regime indicated in Figure 2. The first step was carried out using a vibratory sieve shaker Analysette 3 Pro (Fritsch GmbH, Welden, Germany). This device allows the rapid determination of quantitative particle size distribution and classification by precise sieving with automatic amplitude control according to ISO 9001:2008 [14]. A 2 kg amount of each powder was sieved with the following parameters: initial powder mass 200 g, amplitude 3 mm, working time 5 min, pause 30 s, number of repetitions 5 times. The following fractions, 500 μm, 150 μm, 75 μm, and 45 μm, were obtained for both UHMWPE and salt. Then, the classified powders were mixed using a planetary mill Pulverisetter 5 classic line (Fritsch GmbH, Welden, Germany). It should be mentioned that the materials of the bowls and balls were agate and corundum, respectively. The mixing of powders was carried out in the soft regime: initial powder mass 100 g per bowl and ball size, six balls per bowl, rotating speed of 90 rpm, eight repetitions, working time of 10 min, and pause 2 min to save salt shape and provide powder mixing with the following relation: 90 mass.% of salt to 10 mass.% UHMWPE. Next, the thermal pressing of powders was performed using a hydraulic press Mega KSC-10 A (MEGA Company, Bizkaia, Spain). The final fabrication step was salt dissolution. The compacted polymer-salt composites were dipped into an ultrasonic bath Elmasonic Denta Pro (Elma Schmidbauer GmbH, Singen, Germany) with distilled water for ~30 h, the water was replaced three times during the entire dissolution process.

### 2.2. X-Ray Tomography

X-ray tomography was used to characterize the 3D microstructure and morphological features (thickness, pore shape and size distribution, etc.) of porous UHMWPE samples. A porous cubic bar with size 6 × 6 × 6 mm^3^ was scanned using a Nikon XT H 225 ST CT scanner (Nikon Metrology UK Ltd., Nottingham EMA, UK) with the following settings: X-Ray (170 kV, 270 μA), imaging (exposure 708 ms). Image reconstruction of the radiographs was performed using a back-projection algorithm to produce a stack of 1707 2D slices (1858 × 1889 pixels) with a size of ~20 Gb.

### 2.3. Two- and Three-Dimensional Structure Characterization

The obtained porous structure based on UHMWPE was characterized by both SEM (2D-surface analysis) and X-ray micro-computed tomography with a resolution of 5.7 μm (3D-volume analysis) techniques for initial compression stage. For each dimensional level, the porosity and pore size/shape distributions were estimated.

2D-surface analysis was carried out using the open-source advanced *ImageJ* software [15], namely, the combination of a mean filter with a radius of 10 pixels and a traditional thresholding tool for segmentation, as shown in Figure 3. Statistical information like area, perimeter, circularity and Feret parameters of pores was extracted.

The visualization, cropping, and analysis of the 3D dataset was done using *Avizo* v2020 software (Thermo Fisher Scientific) [16]. The initial dataset was cropped to 400 × 327 × 1707 pixels and then a subregion was extracted 400 × 327 × 850 pixels (2280.5 × 1863.3 × 4852.5 µm). This dataset was reconstructed with a voxel size of 5.7 × 5.7 × 5.7 µm, which was filtered using a median and non-local means filter which were applied in 3D to reduce noise as shown in Figure 4. The material was segmented using interactive thresholding, which binarised the data over a chosen range of intensity. The pore region of the processed data set was extracted by inverting the pixel values and the thickness of the pores was computed using a thickness map module. This technique determines the maximum diameter of a sphere that fits the volume analysed and is based on the work of Hildebrand et al. [17]. The true Euclidean distance metric was computed, and the distance field was also analysed. Finally the 3D computation of connected voxels was performed to extract statistical details such as volume, width, length, 3D Feretshape, etc.

### 2.4. In Situ Compression Test

The compression test was carried out inside the chamber of Scanning Electron Microscope (SEM) Tescan Vega 3 (Tescan, Brno, Czech Republic) using a 1 kN Deben Microtest load cell (Deben Ltd., London, UK). The experiment was performed using a constant displacement rate of 1.5 mm/min, and the system was synchronized with of the SEM image acquisition. A backscattered-electron (BSE) detector was used to collect images at an acceleration voltage of 30 kV, image time of 4 s under low-vacuum (~20 Pa) conditions. Three rectangular bars of the porous UHMWPE (20 × 10 × 10 mm^3^) were compressed according to the ASTM 1621 [18]. The experimental setup is shown in Figure 5.

The typical raw compression curve for the foam recorded by the test device is shown in Figure 6. Young’s modulus was determined from the initial gradient of the plot.

### 2.5. DIC Analysis

The DIC analysis was performed for 83 SEM images acquired during the in-situ compression test using the open-source Ncorr MATLAB package [19]. A region of interest (ROI) was selected such that a significant portion of the porous sample was visible without considering the unstable boundary effects. The DIC parameters subset radius and subset spacing were set to 60 pixels and 3 pixels, respectively. These values were chosen such that Ncorr could best recognize the different regions of foam, and accurately predict seed movement between images. The multithreading option, which corresponds to the number of seeds, was set to three threads. Before the DIC analysis was carried out, the seeds were strategically placed in the ROI for parallel processing. To acquire the ideal seed placement, an iterative approach was taken. Two requirements were considered when relocating seeds during the iterations. The first requirement was that each seed remained within the field of view of all SEM images. The second requirement was that the movement of each seed between consecutive images should not be erratic. The final placement of the seeds is shown in Figure 7.

When formatting the displacements calculated by the program, the unit used was the pixel, since the principal quantity of interest in the subsequent analysis was strain. The strain setting was chosen to obtain the most reliable strain field by further reducing noise in strain data.

After the images were processed by Ncorr, further post processing was carried out separately within the MATLAB environment. Masks were created for selected images through the alteration of contrast and saturation on Adobe Photoshop CC2019, then optimized to reduce speckling and ensure that the material region was large and continuous. These masks were then processed as binary matrices and applied onto the raw displacement data to remove void values. Custom code was written to produce quiver plots from the masked results, where the colour map legend corresponds to the magnitude of the arrow. These quiver plots were then overlaid on the corresponding images. The user interface for the custom post processing program is shown in Figure 8. The strain maps were directly obtained from the Ncorr strain plot, and the colour scale was adjusted to be constant for each image, then inverted. Finally, the masks were then applied to the strain maps to remove the void regions.

## 3. Results and Discussion

### 3.1. Two- and Three-Dimensional Structure Characterisation

The 3D dataset of the porous structure was separated into two phases, material and pores, in order to extract the statistics of the pore characteristics, as shown in Figure 9. Additional 3D graphs of other characteristics can be found in Figure A1.

2D-analysis of SEM images by *ImageJ* showed a porosity value of about ~34.90%, with an average pore width ~150 μm and length ~80 μm, while 3D analysis of X-Ray Tomography data in *Avizo* only reached ~9.17 vol.% porosity and high values of pore characteristics, with width ~100 μm, and length ~220 μm according to the histograms illustrated in Figure 10.

The obtained difference between the 2D and 3D approaches depends on the following issues:(1)3D objects certainly have higher values than 2D due to their geometry and location related to the 2D slice;(2)Segmentation tools are not robust, and strictly depend on the chosen thresholding value in general.

### 3.2. Ashby and Gibson Model

In 1982, Ashby and Gibson proposed the elastic collapse model for three-dimensional cellular materials [20], which correlates the mechanical characteristics of foams with the properties of the cell wall and cell geometry during the test. They predicted the homogenous collapse of the entire porous structure caused by elastic buckling of these members, namely, the elastic collapse of cell walls and struts.

To deeply understand the physical mechanism of polymer foam deformation, this study was separated into two parts for the DIC analysis of the acquired SEM images, for macro and micro scale levels. Macro strain analysis refers to the correlation of the strain obtained from the device with the estimated strain by DIC analysis for the entire field of view of the SEM image, while the micro strain analysis describes the deformation behaviour for several pores with walls and struts, and considering only one pore and its close environment.

### 3.3. Macro Strain Analysis

Figure 11 shows the continuous flow of UHMWPE sponge microstructure during the compression test.

Macro analysis of foam behaviour under compression was obtained from the Ncorr strain data and is displayed in a heatmap configuration in Figure 12. These heat maps indicate how the blue regions of low compressive strain develop into select red areas of high compressive strain greater in magnitude than –0.7. As the foam is compressed, vertical bands of connected regions undergoing high compressive strain become evident, as shown in Figure 12c. The compressive bands arise in a pattern, which links voids that are close together and result from the internal geometry and structure of the foam. The material near a gap can displace into the free space and propagate the high strain regions between voids by cell-edge bending.

### 3.4. Micro Strain Analysis

Analysis of foam behaviour on the micro-scale was conducted in two ways: through displacement vector quiver plots, and strain analysis for the surroundings of a single void. For analysis through displacement data, the obtained Ncorr displacement results were configured with foam masks, and then processed to obtain images showing compression from 1–4 mm for a chosen void. This range of images was selected because it occurs prior to the densification stage and they reveal linearly changing values. The results were displayed as a quiver plot as shown in Figure 13.

The micro-scale compression is visible from observation of the raw foam image, as well as the superimposed displacement vectors from obtained Ncorr data. From image observation alone, the void is expected and observed to become smaller parallel to the X-axis, and to stretch along the Y-axis, with a closing neck region clearly visible in Figure 13d.

This is also supported by the pattern of displacement vectors, whose magnitude is indicated by arrow colour. Global displacement has been factored out to make evident the displacement changes between images. At high compression states prior to densification, material regions pushing on other adjacent material regions will displace less due to the fact that they have fewer degrees of freedom, as they are providing more stability to each other. This is shown in the lower-left region of the central void, where a reduction in light blue arrows to give way to dark blue arrows of a lower magnitude is observed, thus indicating less displacement. The arrows along the top and left of Figure 13a show slight variations in the arrow direction. When compared to the direction variation of the vectors of other images, it is observed that increased compression reduces variation, thus indicating imposed constraints on region movement by increasing compression. Conversely, solid regions next to empty spaces such as voids displace more, as the cell walls are able to bend into the void. The progression towards redder colours in the displacement vectors to the right and top of the central void confirms an increase in displacement magnitude throughout compression, into the void.

The strain analysis for this single void is helpful in displaying the pattern of compression, specifically the way the geometry of the foam affects which regions compress first. The pattern, as seen in Figure 14, is that during compression the length of the void along the X-axis decreases while the length of the void along with the Y-axis increases. The thinnest areas are expected to experience the most strain and close off first. In Figure 14b, the first signs of compression along the void are visible along the cell walls parallel to the Y-axis, concentrated in the thinnest neck region of the void.

This is confirmed through a void area analysis in Figure 15, which shows the reduction in area and how the neck region did indeed close first. As the void continues to compress along the X-axis, the high strain region encompasses more of the cell walls under direct load, while the top and bottom cell walls of the void remain at a lower compressive strain and even link up with the compressive band of a nearby void as shown in Figure 14d. In the final image, the vertical cell walls of the void are clearly under high compressive strain, especially in the closing neck of the void, while the adjacent solid also displays higher compressive strain.

### 3.5. Comparison of DIC Results to Experimental Data

Ncorr has been a very useful tool to qualitatively analyse and understand the behaviour of the foam under compression, but the quantitative data also merits exploration. To obtain macrostrains, the displacement values at the edges of the region of interest were used to calculate the strain in X and strain in Y, as shown in Figure A2.

The point that arises upon the analysis of the plots in Figure 16 is that the DIC-derived transverse Y component of strain begins at zero, but becomes slightly tensile before turning compressive and settling into a steady linear evolution regime. The early anomalous points are likely to be associated with the sample settling in the grips and have been omitted from further analysis for Poisson’s ratio calculation.

To calculate Poisson’s ratio, the strains in the X and Y direction were plotted in Figure 16, lines of best fit calculated, and the negative ratio between the gradients obtained, giving the Poisson’s ratio value of 0.247. The accepted value for the average Poisson’s ratio of UHMWPE from Gibson and Ashby is 0.333. It is expected that the calculated ratio using DIC values would be less than the UHMWPE value presented in literature as DIC only extracts information from one 2D surface plane without considering the geometric and material complexities of the 3D sample.

Figure 17, below, shows the histograms of strain distributions within the sample at different stages of compression. It is apparent that unlike in bulk samples the strain is distributed widely, and that even under overall compression there are some local tensile strains are seen. This highlights the complexity of deformation of porous materials that involves mechanisms such as wall bending, creating steep strain gradients. Furthermore, under small compression in Figure 17a, the strain distribution appears close to Gaussian (normal), although some asymmetry is apparent. Upon further compression, the strain histogram becomes steeper and more asymmetric, so that it is better captured by lognormal distribution. This effect has recently been noted in the work of Chen and Korsunsky (in press) in the context of separation of strain into the elastic part, characterised by normal distribution (that also applies to stress), and plastic strain, which obeys a lognormal distribution. A hypothesis was put forward that this phenomenon is associated with the additive accumulation mechanism that applies to small elastic strains, and multiplicative accumulation mechanism for the larger plastic strain.

The present results illustrate that even under overall elastic deformation that reaches larger values in porous materials considered in the present study, strain histograms tend to evolve from symmetric normal to asymmetric lognormal distributions, as illustrated in Figure 17a–d.

## 4. Conclusions

When the deformation behaviour of porous UHMWPE is studied using modern digital interpretation techniques such as DIC, the observations do not appear to provide a satisfactory match with the Ashby and Gibson theory of 3D cellular materials. The reason for this disagreement is the lack of regularity and repeatability in the response of the random porous structure, e.g., the non-homogeneous collapse of pores observed during compression that was related to spontaneous strain localisation. This highlights the need for introducing a statistical description of deformation inhomogeneity in order to capture the details of macroscale material response.

The 2D experiments on in situ compression of porous UHMWPE foam provide only a superficial view of the deformation mechanisms, and do not capture the phenomena that may be occurring in the bulk material volume. Recognising this limitation of the present study, we plan to run a new set of experiments involving the use of X-Ray Tomography with Digital Volume Correlation (DVC) analysis for porous UHMWPE under in situ compression.

## Figures and Tables

**Figure 1 polymers-12-02607-f001:**
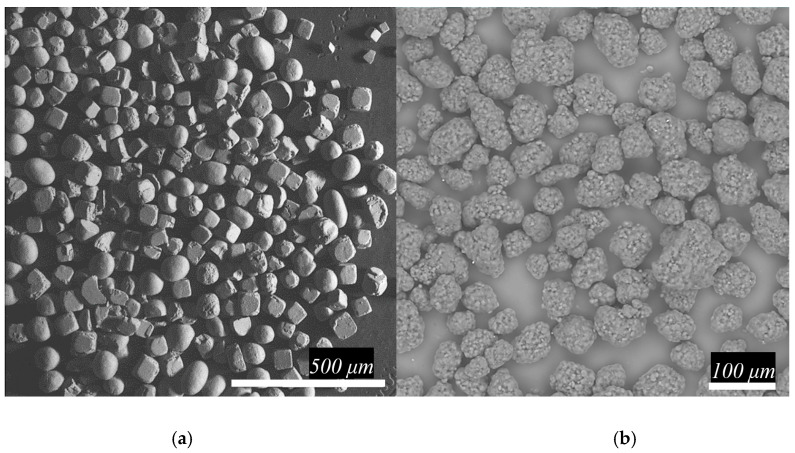
The microphotograph of the used powders: (**a**) table salt and (**b**) UHMWPE.

**Figure 2 polymers-12-02607-f002:**
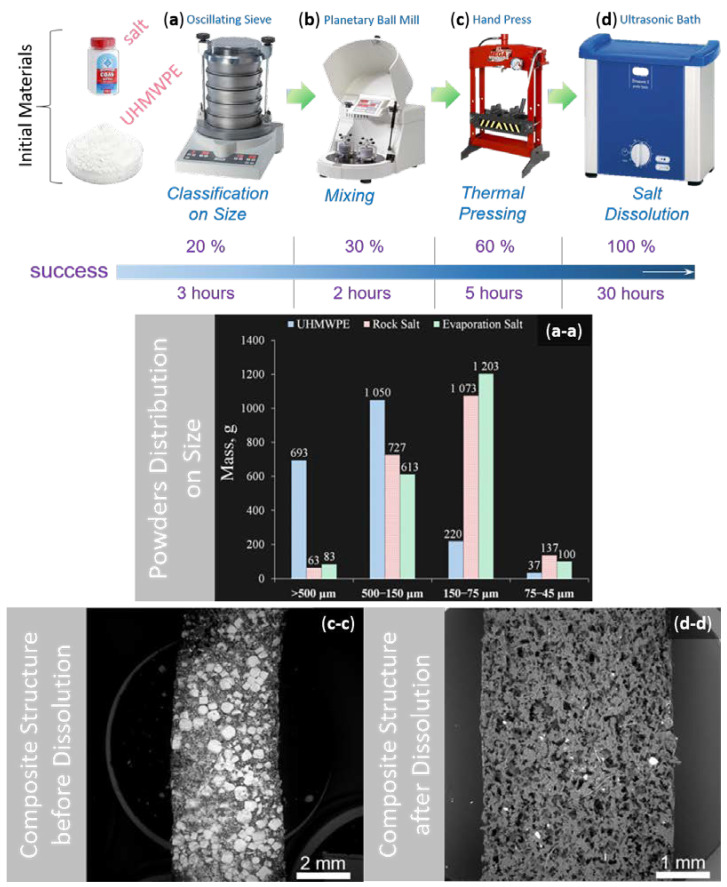
The methodological process for porous sample preparation based on UHMWPE: (**a**) UHMWPE and salt powders classification on size and obtained particles size distributions (**a-a**); (**b**) Mixing of extracting powders in the necessary proportion; (**c**) Thermal pressing of achieved mixture and observed composite structure (**c-c**); (**d**) Desalination process and achieved polymer porous structure (**d-d**).

**Figure 3 polymers-12-02607-f003:**
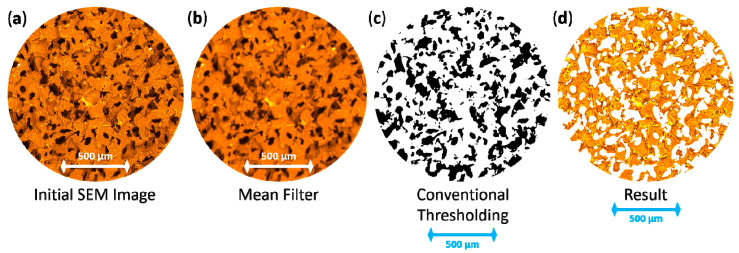
The sequence of SEM image processing (with use of false colour) from raw to segmented data:.(**a**) Initial SEM image of porous structure;.(**b**) Image after mean filter; (**c**) Binary image after traditional thresholding approach; (**d**) Masked initial image.

**Figure 4 polymers-12-02607-f004:**
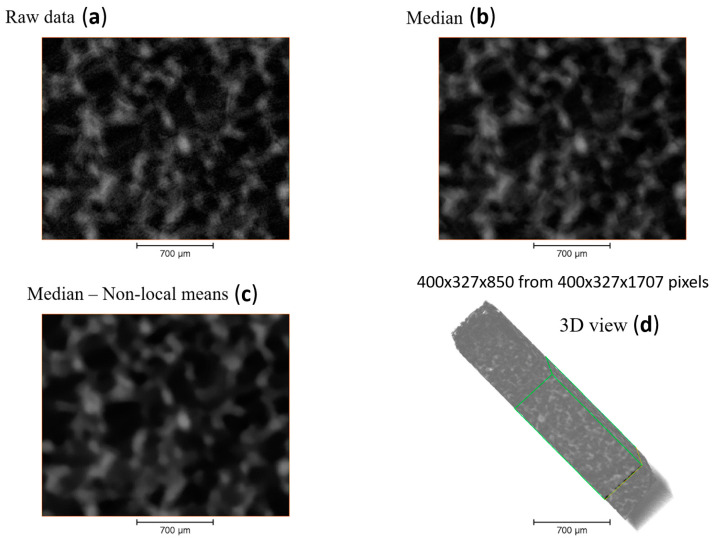
Pre-processing of the obtained 3D dataset: (**a**) raw data; (**b**) data after median filter; (**c**) data after denoising; and (**d**) 3D view of the entire dataset with selected green bounds of the region of interest.

**Figure 5 polymers-12-02607-f005:**
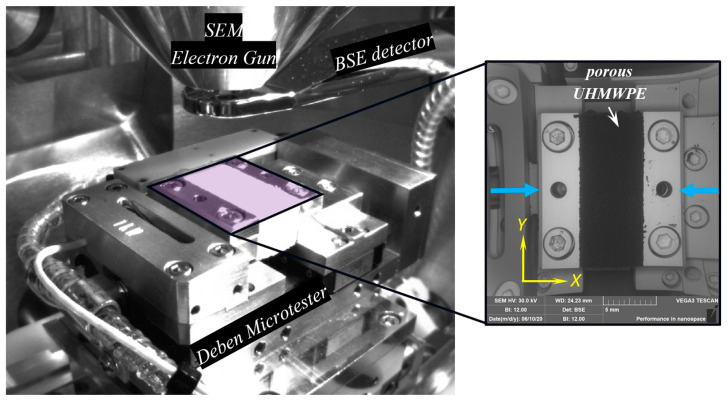
The view field of an experimental setup in SEM.

**Figure 6 polymers-12-02607-f006:**
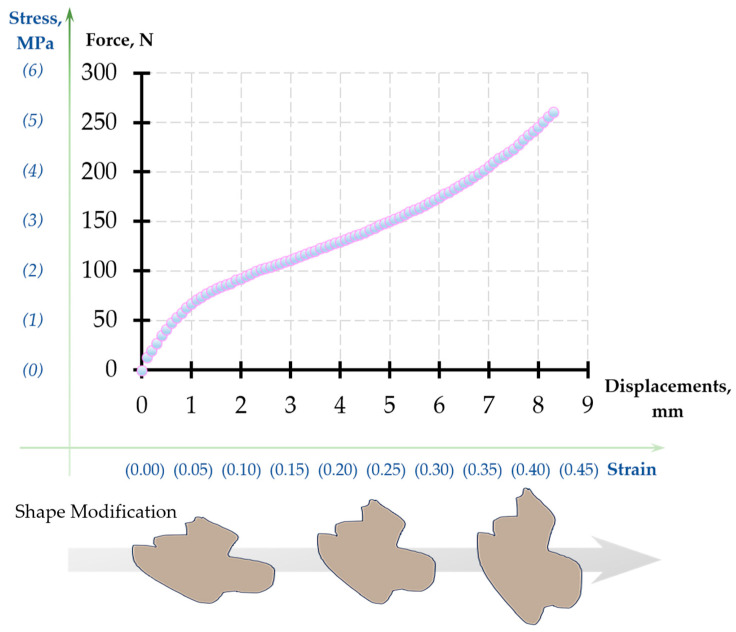
The nominal stress–strain curve obtained from the Deben Microtest for porous UHMWPE.

**Figure 7 polymers-12-02607-f007:**
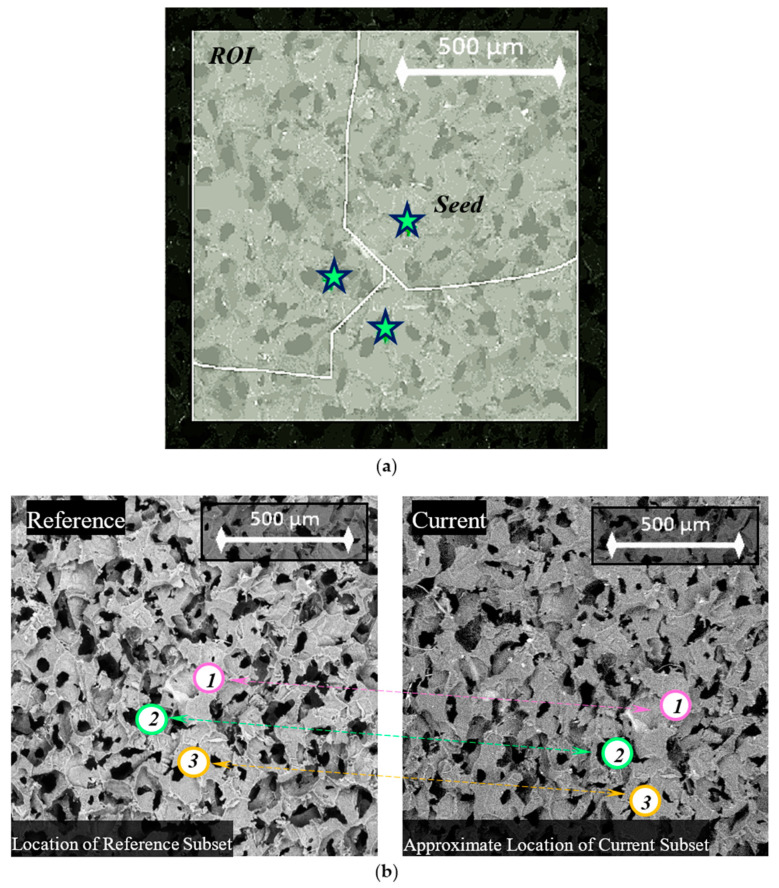
Seed placement on porous sample images captured by SEM: (**a**) clear view of seed regions; (**b**) change in position of seeds from initial to final images.

**Figure 8 polymers-12-02607-f008:**
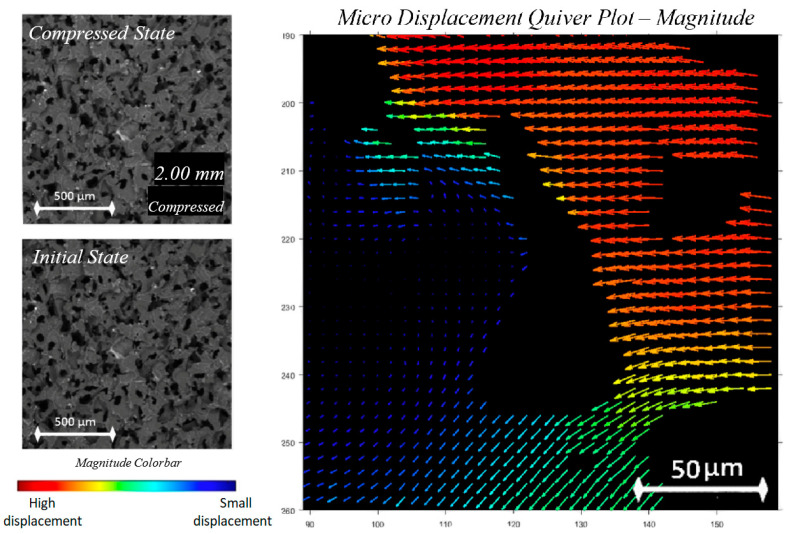
Graphical output from the custom quiver plot code showing magnitude vectors for a select void region, with magnitude and colour corresponding to the rectangular colour bar.

**Figure 9 polymers-12-02607-f009:**
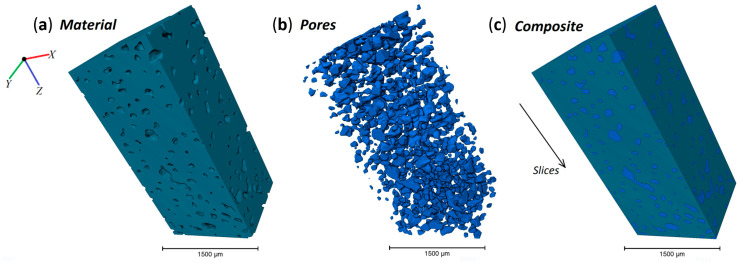
The 3D view of different composite components, namely, (**a**) porous material; (**b**) only pores; and (**c**) material with selected pores.

**Figure 10 polymers-12-02607-f010:**
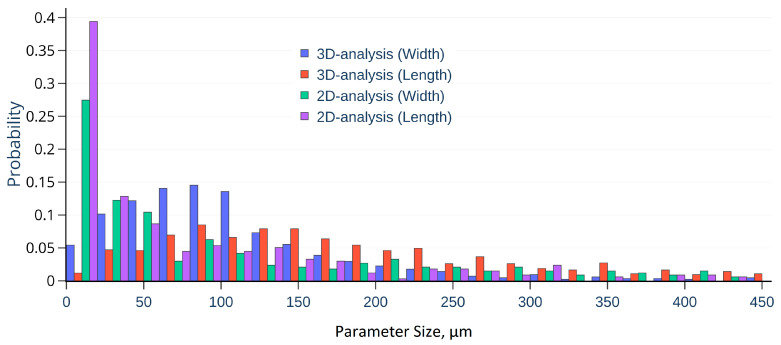
The distribution of pore length and width for 2D and 3D case.

**Figure 11 polymers-12-02607-f011:**
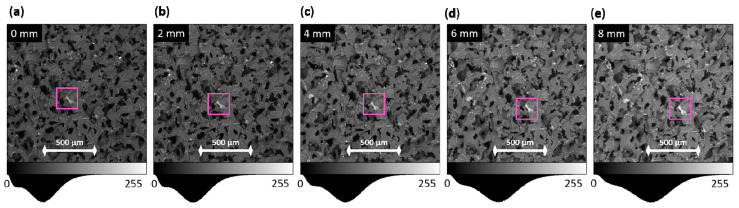
Movement of structural characteristics during compression with plotted intensity distribution histogram below for (**a**) 0 mm; (**b**) 2 mm; (**c**) 4 mm; (**d**) 6 mm; and (**e**) 8 mm compression steps, respectively.

**Figure 12 polymers-12-02607-f012:**
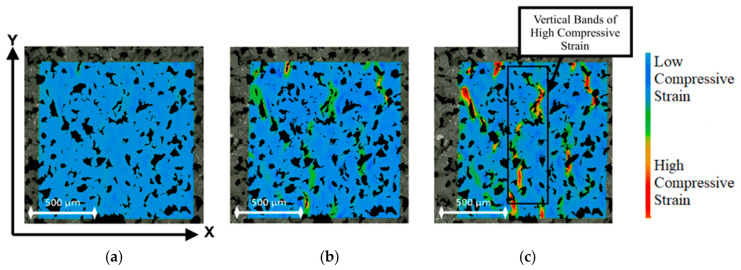
Ncorr strain maps with colour bar for (**a**) 1 mm, (**b**) 2 mm, and (**c**) 3 mm compression.

**Figure 13 polymers-12-02607-f013:**
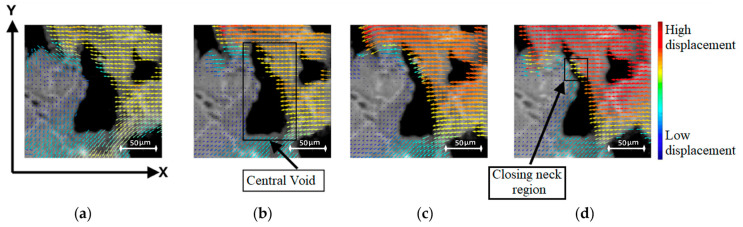
Quiver plot with colour bar showing void compression for (**a**) 1 mm, (**b**) 2 mm, (**c**) 3 mm, and (**d**) 4 mm compression.

**Figure 14 polymers-12-02607-f014:**
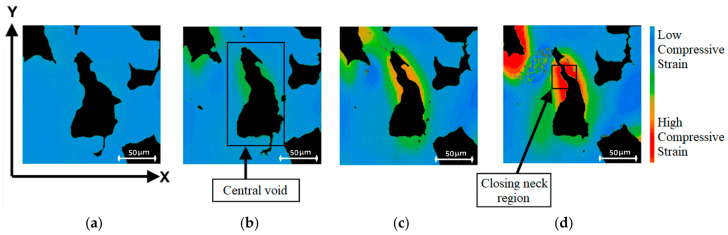
Strain map with colour bar from Ncorr showing strain development for a central void, as seen at (**a**) 1 mm, (**b**) 2 mm, (**c**) 3 mm, and (**d**) 4 mm compression.

**Figure 15 polymers-12-02607-f015:**
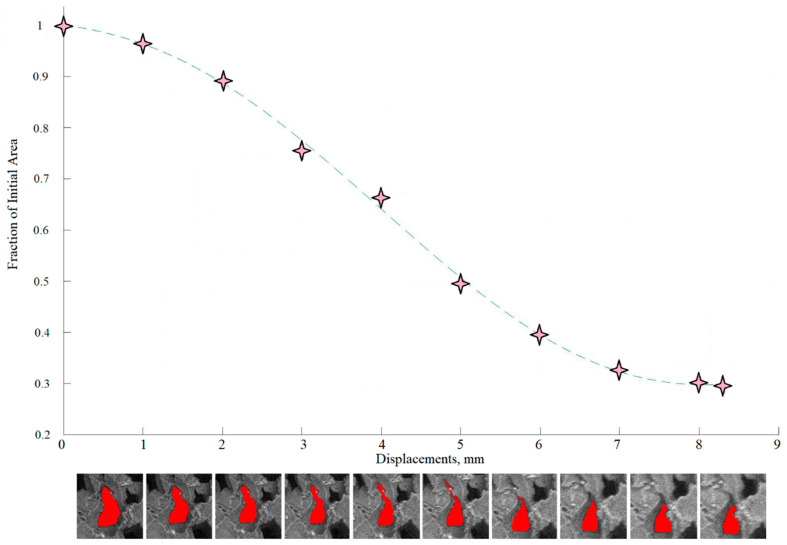
Area progression graph showing qualitative values for the area of a central void during compression.

**Figure 16 polymers-12-02607-f016:**
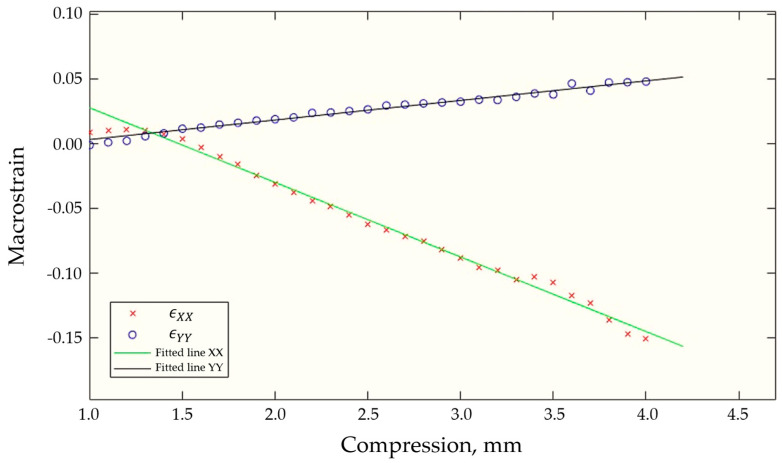
Plot of DIC obtained strain values against compression for analysis and Poisson’s ratio calculation.

**Figure 17 polymers-12-02607-f017:**
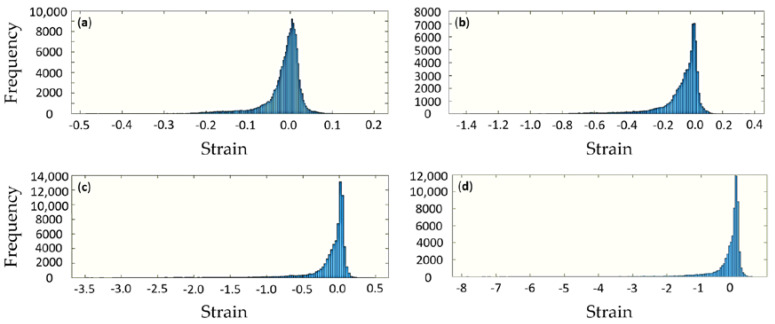
Strain distribution for different steps of compression, namely, nominal displacements: (**a**) 1 mm, (**b**) 2 mm, (**c**) 4 mm, and (**d**) 8 mm, respectively.

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
