# Peer review of "Multi-Scale Digital Image Correlation Analysis of In Situ Deformation of Open-Cell Porous Ultra-High Molecular Weight Polyethylene Foam"

_polymers, 2020, doi:10.3390/polym12112607_

Round 1

Reviewer 1 Report

It is necessary to explain the method to prepare the Porous UHMWPE Sample in present article.

It is suggested that the method should be combined more closely with the specific study topic in this paper, i.e. explainingu the failure model of the  foam structure.  

Author Response

The authors would like to thank the reviewers for the detailed comments and suggestions for the manuscript. We believe that the comments have identified important areas that required improvement. After completion of the suggested edits, the revised manuscript has benefitted from an improvement in the overall presentation and clarity. Below, you will find a point by point description of how each comment was addressed in the manuscript.

Reviewer 2 Report

This paper gives us useful information regarding multi-scale digital image correlation analysis of in situ deformation of open-cell UHMWPE. This manuscript is good paper. However, this manuscript needs some amendment.

  1. The purpose of this paper is too abbreviated. The authors should provide the main findings and the purpose of this paper clearly in introduction part.

The authors should describe the new scientific discovery or importance and originality of this manuscript in detail.

  1. Cell proliferation and cytotoxicity test results should also be presented as the authors mentioned in the abstract.

Author Response

(The authors gave the same response as above.)

Round 2

Reviewer 1 Report

The relationship between digital image and foaming behavior has been elaborated in detail.That's acceptable.

Reviewer 2 Report

I think the manuscript was improved. Some issues remain to be not addressed but majority of the comments were answered. Can be published